# Host-derived protein profiles of human neonatal meconium across gestational ages

Yoshihiko Shitara [1,10], Ryo Konno[2,10], Masahito Yoshihara [3,4,5,10], Kohei Kashima [1], Atsushi Ito[1], Takeo Mukai [1], Goh Kimoto [1], Satsuki Kakiuchi [1], Masaki Ishikawa[2], Tomo Kakihara[6], Takeshi Nagamatsu[7], Naoto Takahashi[1], Jun Fujishiro[6], Eiryo Kawakami [3,4,8], Osamu Ohara [2], Yusuke Kawashima [2] ✉ & Eiichiro Watanabe [6,9] ✉

Meconium, a non-invasive biomaterial reflecting prenatal substance accumulation, could provide valuable insights into neonatal health. However, the comprehensive protein profile of meconium across gestational ages remains unclear. Here, we conducted an extensive proteomic analysis of first meconium from 259 newborns across varied gestational ages to delineate protein composition and elucidate its relevance to neonatal diseases. The first meconium samples were collected, with the majority obtained before feeding, and the mean time for the first meconium passage from the anus was 11.9 ± 9.47 h. Our analysis revealed 5370 host-derived meconium proteins, which varied depending on sex and gestational age. Specifically, meconium from preterm infants exhibited elevated concentrations of proteins associated with the extracellular matrix. Additionally, the protein profiles of meconium also exhibited unique variations depending on both specific diseases, including gastrointestinal diseases, congenital heart diseases, and maternal conditions. Furthermore, we developed a machine learning model to predict gestational ages using meconium proteins. Our model suggests that newborns with gastrointestinal diseases and congenital heart diseases may have immature gastrointestinal systems. These findings highlight the intricate relationship between clinical parameters and meconium protein composition, offering potential for a novel approach to assess neonatal gastrointestinal health.

Human viability, which refers to the gestational age at which the chance of survival is 50%, is currently estimated at 23–24 weeks in developed countries. Despite significant technological advancements and the efforts of child health experts, extremely preterm infants (those with less than 28 weeks of gestation) and extremely low birth weight infants (those whose weight is less than 1000 g) continue to face high risks of mortality and disability[1–3]. Moreover, these infants are vulnerable to life-threatening surgical intestinal disorders such as

[1]Department of Pediatrics, Faculty of Medicine, The University of Tokyo, Tokyo, Japan. [2]Department of Applied Genomics, Kazusa DNA Research Institute, Chiba, Japan. [3]Institute for Advanced Academic Research (IAAR), Chiba University, Chiba, Japan. [4]Department of Artificial Intelligence Medicine, Graduate School of Medicine, Chiba University, Chiba, Japan. [5]Premium Research Institute for Human Metaverse Medicine (WPI-PRIMe), Osaka University, Suita, Osaka, Japan. [6]Department of Pediatric Surgery, Faculty of Medicine, The University of Tokyo, Tokyo, Japan. [7]Department of Obstetrics and Gynecology, Faculty of Medicine, International University of Health and Welfare, Chiba, Japan. [8]Advanced Data Science Project, RIKEN Information R&D and Strategy Headquarters, RIKEN, Kanagawa, Japan. [9]Department of Surgery, Gunma Children's Medical Center, Gunma, Japan. [10]These authors contributed equally: Yoshihiko Shitara, Ryo Konno, Masahito Yoshihara. ✉e-mail: ykawashi@kazusa.or.jp; eiichiro.watanabe.riken@gmail.com

necrotizing enterocolitis (NEC) and meconium-related ileus (MRI)[2,4,5]. Despite numerous studies investigating the blood, urine, and tissues of these infants[5,6], the underlying mechanisms of these intestinal disorders remain unclear, and diagnostic biomarkers are not yet available.

Proteome analysis enables the comprehensive analysis of protein composition in various body fluids and tissues[7–10] and has been widely applied in numerous clinical studies in newborn babies[11–15]. In recent years, mass spectrometry has undergone significant advancements, enabling the detection of over 10,000 proteins within a single run in cellular or tissue samples[16–18]. Utilizing this method, we previously identified over 2000 proteins in stool proteome analysis[19]. Additionally, the amniotic fluid proteome has been used to elucidate the intrauterine environment associated with sex, gestational age, and specific diseases, such as gastrointestinal diseases (GID), congenital heart diseases (CHD), chromosomal abnormalities (CA), and congenital infection diseases (CID)[20–25].

Meconium is a viscid, odourless, greenish-black material present in the foetal intestine from approximately the 12th week of gestation. In full-term neonates, the initial meconium is typically passed within 48 h after birth. Thus, it can be collected noninvasively after birth, making it an ideal clinical sample for investigating the gastrointestinal pathophysiological condition of newborn babies[26–30].

Because meconium is minimally excreted by the foetus during gestation but is defaecated after birth, its components accumulate exclusively during intrauterine life[31]. The composition of the meconium may vary depending on the time of its formation in the foetal intestine. Therefore, it is believed that the composition of the initial meconium differs from that of the subsequent stool passed during the postnatal period[32,33]. In addition, recent research indicates that in cases where a foetus develops without any abnormal maternal conditions, the meconium does not harbour the gut microbiome prior to birth[34]. Therefore, early meconium seems to be minimally influenced by microbial interactions. Protein profile analysis of the early meconium has the potential to elucidate the gastrointestinal environment of newborns and may provide information on laboratory parameters that describe the intrauterine environment[35–37].

We hypothesized that employing proteomic analysis of meconium could offer a novel approach to both identify stool biomarkers and uncover previously unknown aetiologies linked to neonatal diseases. In this study, we aimed to establish human-derived protein profiles of the initial meconium across varied gestational ages, identify meconium biomarkers, and uncover the underlying causes of disease in newborns through deep proteome analyses.

## Results

### Comprehensive proteomic analysis reveals abundant host-derived proteins from diverse tissues condensed in the meconium

We performed a data-independent acquisition (DIA) proteomic analysis—a highly effective method for quantifying proteins in various human-derived sources in clinical research[7–10]—of initial meconium samples collected from 259 newborns (Fig. 1a). The majority (76.4%) of the first meconium samples used in this study were collected before feeding, while a portion (23.6%) were obtained after feeding. The mean time for the first meconium passage from the anus was 11.9 ± 9.47 h. In total, 259 newborns, including 163 preterm infants, who received care in the Neonatal Intensive Care Unit (NICU) or nursery room at the University of Tokyo Hospital were included (Table 1, Table 2, Fig. 1b, and Supplementary Table 1). In this study, the DIA-proteome analysis successfully quantified 5370 human-derived proteins interspersed in the first meconium of the 259 newborns (Fig. 1c). We observed that among the 5370 proteins identified, some were only detected in a limited number of samples. Consequently, we opted to focus on the 3433 proteins that were detected in at least 100 samples, and these were used for subsequent analyses (Fig. 1c). Notably, our findings

demonstrated that the first meconium contains a diverse array of human-derived proteins not only from the gastrointestinal tract but also from various tissues (Fig. 1d) that have numerous biological functions (Fig. 1e). Therefore, DIA-proteome analysis holds immense potential as a method for identifying biomarkers and shedding light on the elusive aetiology of diseases in newborns through human-derived proteins within the meconium.

### Divergent meconium protein signatures between the sexes

Reports indicate differences in gene expression and protein synthesis between males and females, and these differences have been attributed to DNA methylation of the whole genome resulting from X chromosome inactivation[38–40].

In neonatology, female newborns exhibit greater tolerance to GID, such as NEC and MRI than males, especially in preterm infants[41–43]. Therefore, we investigated the potential association between sex and meconium protein profiles. We identified 77 proteins that exhibited a significant difference between the sexes ($P < 0.05$). Of these, 67 proteins were more abundant in females, while only 10 were more abundant in males (Fig. 2a and Supplementary Data 1). Notably, proteins such as Transmembrane protease serine 11D (TMPRSS11D), Serpin B4 (SERPINB4), Desmoglein-3 (DSG3), Short-chain dehydrogenase/reductase family 9C member 7 (SDR9C7), Cystatin-A (CSTA), Serpin B3 (SERPINB3), Cystatin-B (CSTB), Uridine phosphorylase 1 (UPP1), Transmembrane protease serine 11E (TMPRSS11E), and Serpin B13 (SERPINB13) were more abundant in female than in male newborns, whereas adhesion G protein-coupled receptor L1 (ADGRL1) was more abundant in male than in female newborns. These proteins were consistently more abundant in females or males in an independent cohort comprising 79 non-diseased newborns (Supplementary Fig. 1).

To estimate the origins of these sex-dependent proteins, we first conducted a gene set enrichment analysis (GSEA) using the list of tissue-specific proteins as gene sets. Proteins more abundant in females were found to be significantly enriched in squamous epithelial tissues such as the oesophagus, vagina, skin, and cervix (Fig. 2b). These proteins, abundant in females, are hypothesized to originate from the vagina, considering that during embryonic development, the rectum, vagina, and urethra form separately from the common channel known as the cloaca[44]. Next, GSEA was conducted to explore the biological processes enriched in the sex-dependent proteins. Regulation of endopeptidase activity, regulation of peptidase activity, negative regulation of peptidase activity, negative regulation of hydrolase activity, negative regulation of endopeptidase activity, negative regulation of proteolysis, epidermis development, humoral immune response, regulation of proteolysis, and cell-cell adhesion were the top ten gene ontology (GO) terms in females. In contrast, small GTPase mediated signal transduction, microtubule-based process, regulation of cellular component biogenesis, organelle assembly, mitotic cell cycle process, organelle fission, mitotic cell cycle, nuclear division, cell cycle, and mitotic nuclear division were the top ten GO terms in males (Fig. 2c). In females, proteins associated with immune responses, including humoral immune response, were enriched. These differences in protein profiles within the meconium between the sexes could potentially impact the stress tolerance of the gastrointestinal tract during early postnatal development.

### Insights into protein profiles across varied gestational ages, extracellular matrix proteins and mucin profiles in preterm gestational neonates

We explored the potential association between gestational age (GA) and meconium protein profiles, as GA correlates with meconium characteristics such as viscosity[45], mineral components[46], and glycoprotein composition[47]. First, we utilized a simple linear fitting approach to analyse changes in protein levels across GAs (Supplementary Fig. 2a). A total of 1158 proteins were found to significantly decrease, whereas 348 proteins exhibited an increase with GA.

**Fig. 1 | Data-independent acquisition proteome analysis of the first meconium.**
**a** Schematic overview of the analytical process used for the comprehensive analysis of the meconium proteome. W weeks. **b** Flowchart of the newborn baby selection process. NICU neonatal intensive care unit. **c** Detection of human-derived meconium proteins using proteomic analysis. The x-axis represents the number of samples in which the number of proteins shown on the y-axis was detected.
**d** Verification of the origin of the identified 3433 proteins using the Human Protein Atlas (HPA). The bar plot displays the number of tissue-specific proteins, while the line graph shows the proportion of proteins identified in each tissue. Tissues are sorted based on the proportion of proteins. **e** Gene ontology (GO) term over-representation analysis of the identified 3433 proteins. The bar plot displays the number of proteins associated with each category. Colours represent the significance of enrichment. The top 20 significantly enriched GO terms are shown. *P*-values were calculated using a one-sided hypergeometric test with Benjamini–Hochberg correction.

**Table 1 | Detailed clinical neonatal characteristics of study participants**

|  | N (%) | Mean (SD) | Median (IQR) |
|---|---|---|---|
| *Neonatal characteristics* |  |  |  |
| Gestational age at birth |  | 34.9 (4.1) |  |
| < 28 weeks | 20 (7.7) |  |  |
| 28–32 weeks | 29 (11.2) |  |  |
| 32–37 weeks | 114 (44.0) |  |  |
| ≥ 37 weeks | 96 (37.1) |  |  |
| Birth weight |  | 2132.6 (777.8) |  |
| <1000 g | 23 (8.9) |  |  |
| 1000–1500 g | 25 (9.7) |  |  |
| 1500–2500 g | 131 (50.6) |  |  |
| 2500–4000 g | 78 (30.1) |  |  |
| ≥ 4000 g | 2 (0.8) |  |  |
| Birth weight SD score |  | − 0.45 (1.25) |  |
| Birth length (cm) |  | 42.7 (5.35) |  |
| Birth length SD score |  | − 0.78 (1.07) |  |
| Head circumference (cm) |  | 30.7 (3.36) |  |
| Head circumference SD score |  | − 0.08 (0.94) |  |
| Sex, male | 112 (43.2) |  |  |
| Small for gestational age, SGA | 62 (23.9) |  |  |
| Heavy for gestational age, HGA | 22 (8.5) |  |  |
| Apgar score 1 |  |  | 8 (7, 8) |
| Apgar score 5 |  |  | 9 (8, 9) |
| UApH[*] |  | 7.27 (0.07) |  |

*UA umbilical artery, NA not available, SD standard deviation, IQR interquartile range*
[*]NA = 25

**Table 2 | Detailed clinical maternal characteristics of study participants**

|  | N (%) | Mean (SD) |
|---|---|---|
| *Maternal characteristics* |  |  |
| Maternal age (years) |  | 34.8 (5.1) |
| Primipara | 144 (55.6) |  |
| Natural conception | 180 (69.5) |  |
| Singleton birth | 211 (81.5) |  |
| Delivery mode (vaginal delivery) | 80 (30.9) |  |
| Premature rupture of the membranes, PROM | 46 (17.8) |  |
| Chorioamnionitis, CAM[*a] | 51 (24.1) |  |
| Clinical CAM | 6 (2.3) |  |
| Funisitis[*b] | 28 (13.2) |  |
| Intrauterine infection | 11 (4.2) |  |
| Gestational diabetes mellitus, GDM | 23 (8.9) |  |
| Hypertensive disorder of pregnancy, HDP | 22 (8.5) |  |
| Hypertension | 2 (0.8) |  |
| Antenatal corticosteroids | 69 (26.6) |  |
| Alcohol drinking before pregnancy[*c] | 10 (4.0) |  |
| Smoking before pregnancy[*d] | 1 (0.4) |  |
| Smoking during pregnancy[*e] | 8 (3.3) |  |

*NA not available, SD standard deviation*
[*a]NA = 47
[*b]NA = 47
[*c]NA = 15
[*d]NA = 17
[*e]NA = 16

We further employed the locally estimated scatterplot smoothing (LOESS) regression model[48] to investigate undulating, non-linear protein trajectories in relation to GA, as were observed in the ageing patterns of proteins in the plasma and aqueous humor[48,49] (Supplementary Fig. 2b). Using unsupervised hierarchical clustering, the proteins were grouped into six clusters according to their trajectory patterns (Fig. 3a, b, and Supplementary Data 2). Clusters 1 and 2 showed an increasing pattern in the early gestational stage, both of which were enriched with plasma membrane organization, membrane fission, or mitotic cytokinesis, suggesting the active processes related to cell growth, division, and repair at this stage (Fig. 3c). Notably, the presence of alkaline phosphatase families, including intestinal-type alkaline phosphatase, placental-type alkaline phosphatase, and germ cell-type alkaline phosphatase, within these clusters aligns with a prior study highlighting a significant positive association between meconium alkaline phosphatase activity and GA[50]. Additionally, cluster 4 showed a unique pattern, monotonically decreasing with GA. These proteins were specifically enriched both with extracellular matrix (ECM) and extracellular structure-related proteins (Fig. 3c and Supplementary Data 3). Notably, proteins such as laminin subunit gamma-1 (LAMC1), integrin alpha-1 (ITGA1), integrin alpha-2 (ITGA2), integrin beta-1 (ITGB1), collagen VI alpha-3 chain (COL6A3), and collagen I alpha-1 chain (COL1A1) were remarkably abundant in the meconium of preterm gestational (PG) infants (Supplementary Fig. 2c). Furthermore, these findings were corroborated in an independent cohort comprising 79 newborns (Supplementary Fig. 2d). During the embryonic stages, ECM proteins such as laminin and integrin were expressed in both the villi and crypts along the gastrointestinal tract[51]. Therefore, these findings were considered as reflections of the course of development in the embryonic stages.

Meconium, a viscous excrement, can experience delayed passage owing to increased viscosity[26]. In addition to the physiologically immature motility of the intestine, the reduced water content in the stool of premature babies can contribute to MRI because of increased mucoviscidosis[27,52]. Mucus, a gel-like material found on mucosal surfaces, predominantly consists of one or more gel-forming mucins, and is considered a potential contributor to MRI[53]. However, a comprehensive mucin profile of the meconium remains elusive. Our proteomic analysis identified multiple mucins in the first meconium. Specifically, Mucin-1, Mucin-2, Mucin-3A, Mucin-6, and Mucin-7 were grouped in cluster 1, Mucin-5AC and Mucin-16 were in cluster 4, Mucin-4 and Mucin-17 were in cluster 5, while Mucin-5B, Mucin-12, and Mucin-13 were assigned to cluster 6 (Supplementary Table 2). Among them, Mucin-2, primarily synthesized in the small and large intestine, is extensively studied due to its association with intestinal disorders, such as inflammatory bowel diseases, when deficient[54,55]. To validate our proteome dataset, we employed an enzyme-linked immunosorbent assay (ELISA) tailored for the precise quantification of individual meconium proteins. Our ELISA analysis of initial meconium samples revealed a notable alignment of Mucin-2 levels with proteomic data, a correlation further confirmed across separate cohorts (Supplementary Fig. 3a). These ELISA findings offer valuable insights into the expression pattern of Mucin-2 and its potential relevance in neonates across different gestational ages.

In summary, the meconium of PG infants displayed condensed proteins linked with the ECM, along with a distinct mucin profile characterized by low levels of Mucin-1 and Mucin-2, and elevated levels of Mucin-5AC and Mucin-16 (Supplementary Fig. 3b). These unique meconium profiles are influenced by infant maturity and could potentially play a role in the pathogenesis of serious intestinal disorders commonly observed in newborns during the early postnatal period.

### Disease-dependent meconium protein profiles
We aimed to uncover potential associations between diseases and the meconium proteome to shed light on the underlying pathogenesis of various diseases such as GID, CHD, CA, and CID[20–25] (Supplementary Fig. 4a, Supplementary Data 4). As a result, distinct meconium protein

**a**

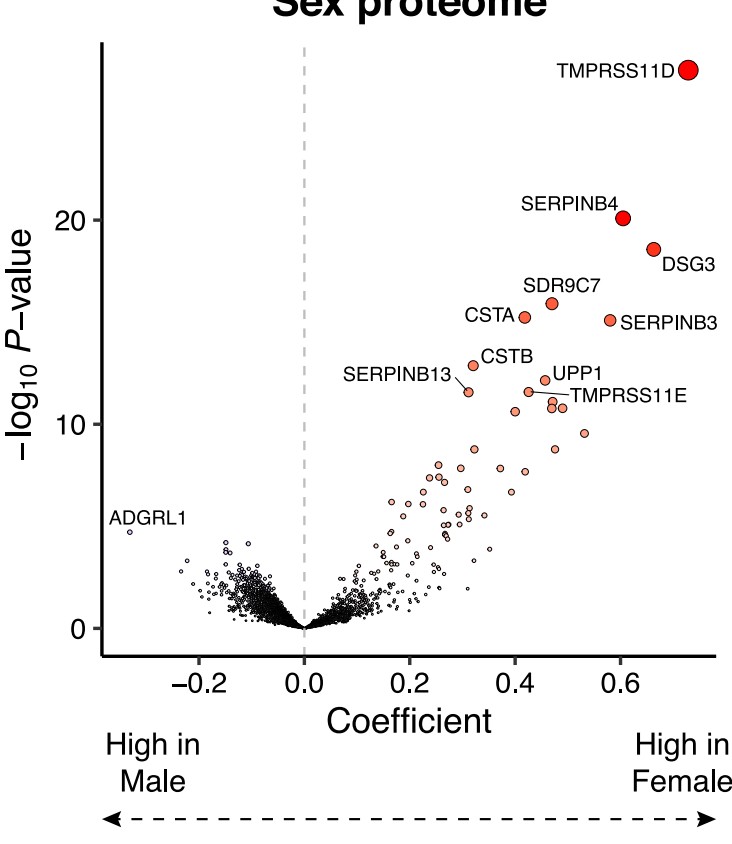

**b**

**c**

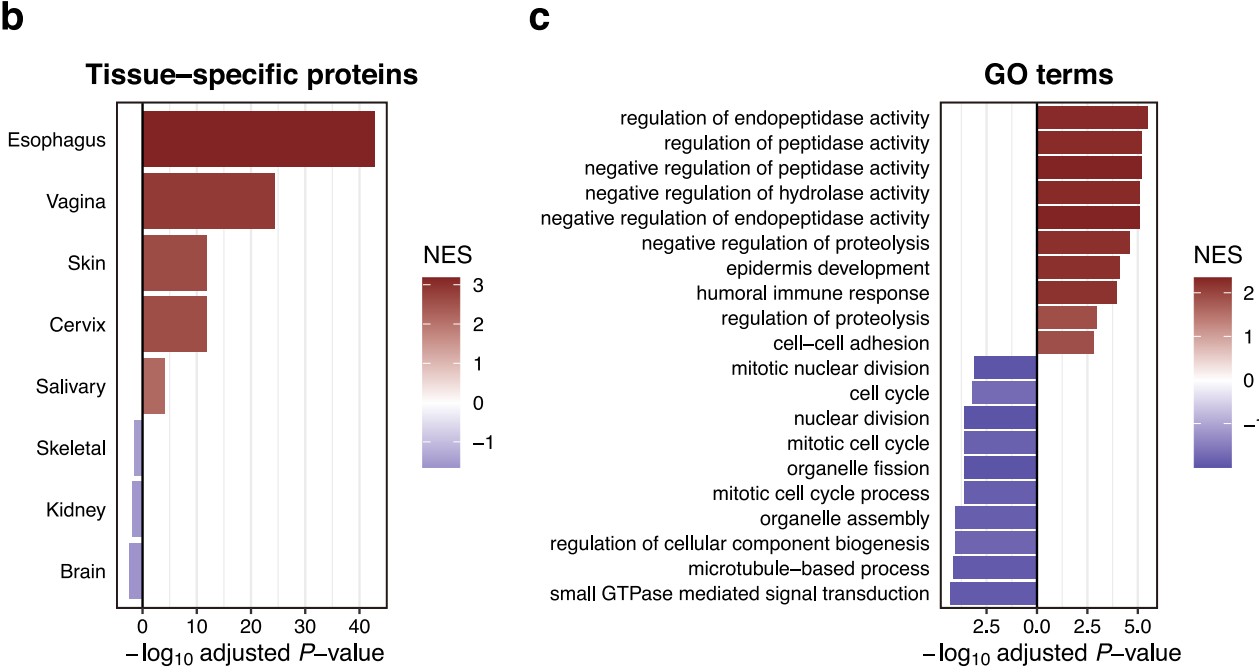

profiles were identified for each disease category (Fig. 4a–d). In the GID cohort, we observed enrichment in metabolic processes, including catabolic processes, which were not evident in the other three disease groups (Fig. 4e). Additionally, Ras protein signal transduction, vital for intestinal epithelial maturation[56], is significantly downregulated in the GID cohort. This reduction in Ras protein function likely contributes to aberrant gastrointestinal organization. Consequently, anatomical

anomalies linked to GID may potentially affect the functional roles of meconium proteins. In the CHD group, there was a decrease observed in proteins linked to localisation within membranes, accompanied by diminished organelle localisation (Fig. 4f). In the CA group, proteins associated with microtubule-based processes and the organization of the microtubule cytoskeleton showed diminished levels (Fig. 4g). Within our CA cohorts, Trisomy 21 was notably prevalent

**Fig. 2 | Differential composition of host-derived meconium proteins between males and females. a** Volcano plot showing the changes in the meconium proteome between males and females. The x-axis represents the effect size, indicated by the coefficient, while the y-axis displays the statistical significance, represented by -$\log_{10}$(P-value). Positive coefficients indicate higher abundance in females, while negative coefficients indicate higher abundance in males. P-values for each coefficient were calculated using a two-sided t-test without adjustment for multiple testing. **b** Bar plots showing the significant enrichment (adjusted $P < 0.05$) of tissues specifically expressing proteins that are abundant in females (red) and males (blue). P-values were calculated using a permutation test with Benjamini−Hochberg correction, based on the GSEA algorithm as implemented in the R package fgsea. **c** Bar plots showing the top 10 significantly enriched Gene ontology (GO) terms of proteins that are abundant in females (red) and males (blue). NES normalised enrichment score. P-values were calculated using a permutation test with Benjamini−Hochberg correction, based on the GSEA algorithm as implemented in the R package fgsea.

(Supplementary Data 4), marked by distinctive features such as ciliopathies, where microtubules played a crucial role[57]. The composition of meconium in these cohorts may serve as a reflection of these clinical characteristics. Moreover, in the CID cohort, the top ten upregulated enrichment processes were associated with responses to infection (Fig. 4h). Previous studies have reported differences in the CID proteomic composition of amniotic fluid compared to normal conditions (25), and it is well-documented that infants ingest amniotic fluid, consequently influencing meconium composition. This finding underscores the direct impact of amniotic fluid on meconium composition. Furthermore, we conducted an analysis of the meconium proteome in cases of gestational diabetes mellitus (GDM) and hypertensive disorders of pregnancy (HDP). Our findings revealed enrichment of transport functions in babies born to mothers with GDM, whereas babies born to mothers with HDP exhibited enrichment of biosynthesis and metabolism functions (Supplementary Fig. 4b, c). Building upon our study's findings regarding meconium, there is potential to expand our investigation towards disease prediction by utilizing samples of amniotic fluid. Collectively, these unique meconium protein profiles could potentially serve as indicators of pathogenesis for specific neonatal diseases, reflecting intrauterine conditions.

### Potential for GA prediction through meconium proteome analysis

To ascertain whether the composition of meconium proteins could serve as a predictor for GA, we applied a least absolute shrinkage and selection operator (LASSO) regression. First, we constructed and validated our model using a training and validation cohort approach, randomly dividing the 259 samples at a 2:1 ratio to develop an initial predictive gestational model (Supplementary Fig. 5a). However, we encountered discrepancies between our predictions and the actual samples, which could be attributed to the effect of certain diseases. As the meconium samples obtained from the GID, CHD, CA, and CID cohorts had unique protein profiles (refer to Fig. 4), we refined our GA prediction model by excluding samples afflicted with these four specific diseases. The samples were divided into three cohorts: a training cohort of 149 non-disease samples (excluding those with specific diseases), a validation cohort of 55 non-disease samples, and another validation cohort of 55 specific-disease samples. The novel GA prediction model was developed based on 57 proteins (Supplementary Data 5) using the training cohort data ($R = 0.98$, RMSE = 1.00; Fig. 5a). This model demonstrated high accuracy in predicting the GA of the validation cohort of the non-disease samples ($R = 0.93$, RMSE = 1.36; Fig. 5b), but low accuracy in predicting the GA of the validation cohort of the specific-disease samples ($R = 0.81$, RMSE = 2.57; Fig. 5c). Notably, there were outliers in the validation cohort comprising the 55 specific-disease samples, where the predicted ages were lower than the actual ages. These outliers were disproportionately represented in samples with GID and CHD (Supplementary Fig. 5b). These GID and CHD samples showed significantly lower GA discrepancies (predicted GA − actual GA) compared to the non-disease samples ($P = 0.002$ and 0.0002, respectively; Fig. 5d), suggesting that newborns in the GID cohort may experience gastrointestinal immaturity, while growth retardation in the CHD cohort appears to be secondary to abnormalities in blood flow and oxygenation[58]. Finally, we successfully validated our prediction model by assessing an external cohort comprising 79 non-diseased newborns, confirming its effectiveness in estimating GA ($R = 0.88$, RMSE = 2.24; Supplementary Fig. 5c).

## Discussion

Here we conducted an extensive proteomic analysis of meconium from 259 newborns to establish a human-derived protein profile and elucidate its relevance to neonatal diseases. Our analysis revealed 5370 host-derived meconium proteins and demonstrated variation depending on sex and GA by focusing on 3433 proteins. Additionally, similar to proteomic studies conducted on amniotic fluids, the protein profiles of meconium also exhibited unique variations depending on both specific diseases, including GID, CHD, CA, and CID, and maternal conditions, such as GDM and HDP. These findings highlight the intricate relationship between clinical parameters and meconium protein composition, offering potential for a novel approach to assess neonatal gastrointestinal health. Furthermore, we developed a machine learning model to predict GA using meconium proteins. While ultrasound dating remains the gold standard for assessing GA in early pregnancy, its accuracy poses challenges in low- and middle-income countries[59]. In these contexts, our innovative approach leveraging meconium proteome analysis offers a promising solution. By accurately estimating GA, it can inform post-birth medical interventions and elevate global public health standards.

This study has several limitations. First, the sample size was limited, particularly for diseases such as NEC and MRI. Furthermore, there was a scarcity of PG samples within the specific-disease cohort, and the study was confined to a single institution. Moreover, in the case of the chorioamnionitis (CAM) cohort, we were unable to include these samples owing to a high rate of missing values. Consequently, our ability to investigate potential connections between meconium proteins and these diseases was restricted. Nevertheless, we anticipate that larger-scale cohort studies will reveal specific meconium protein biomarkers associated with these critical neonatal conditions. Second, our study focused solely on meconium proteins, overlooking other critical clinical proteomes such as those of the amniotic fluid and plasma. Although collecting meconium during pregnancy for foetal condition assessment is impractical, recent reports indicate trace amounts of meconium in foetal amniotic fluid[31]. Establishing a correlation between the proteomics of amniotic fluid and meconium could lay the groundwork for future research. This revelation hints at the possibility of harnessing human meconium insights to develop biomarkers through the amniocentesis approach. Thirdly, the timing of sampling the initial meconium differs between our cohorts, and it may be necessary to validate our findings using more sophisticated methods. A previous study has suggested that, in infant gut microbiota research, rectal swab sampling could serve as a reliable alternative to sampling via natural defecation[60]. Although there are no prior reports specifically addressing sampling methods in meconium research, we believe that conducting studies on sampling methodologies while prioritizing infant safety could help address this issue. Finally, shotgun proteome analysis identifies a protein by detecting specific peptides derived from it. Hence, a single proteome analysis does not conclusively determine the presence of an entire protein in the samples or its functions. However, shotgun proteome analysis remains a superior method for detecting biomarkers, as it can detect short peptides in the

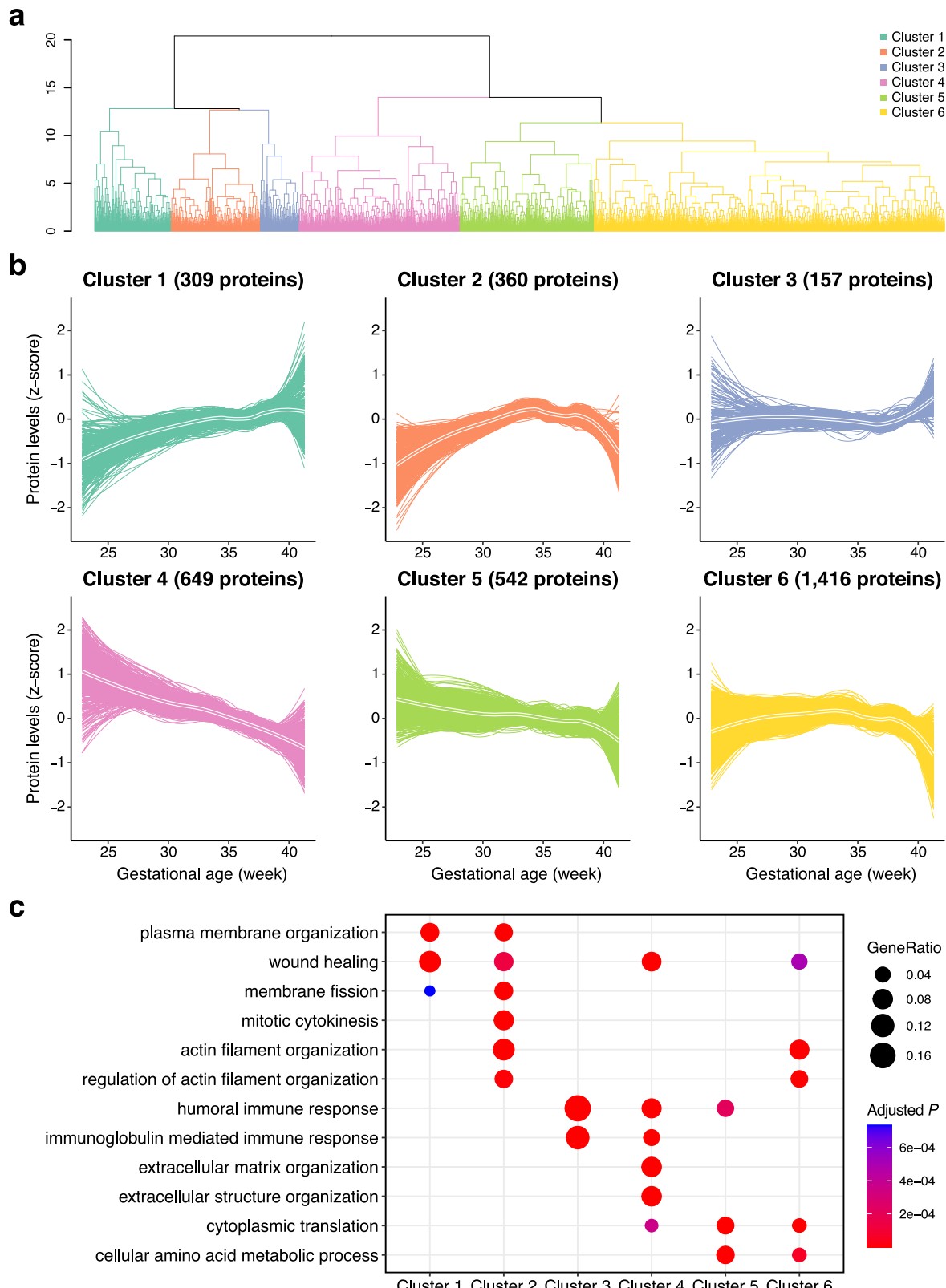

**Fig. 3 | Clustering of host-derived meconium protein trajectories during gestational ageing. a** Unsupervised hierarchical clustering of 3,433 meconium proteins with similar trajectories. **b** Protein trajectories of the six identified clusters. Thicker lines represent the average trajectory for each cluster. **c** Gene ontology (GO) term enrichment analysis of the proteins in each cluster. *P*-values were calculated using a one-sided hypergeometric test with Benjamini–Hochberg correction.

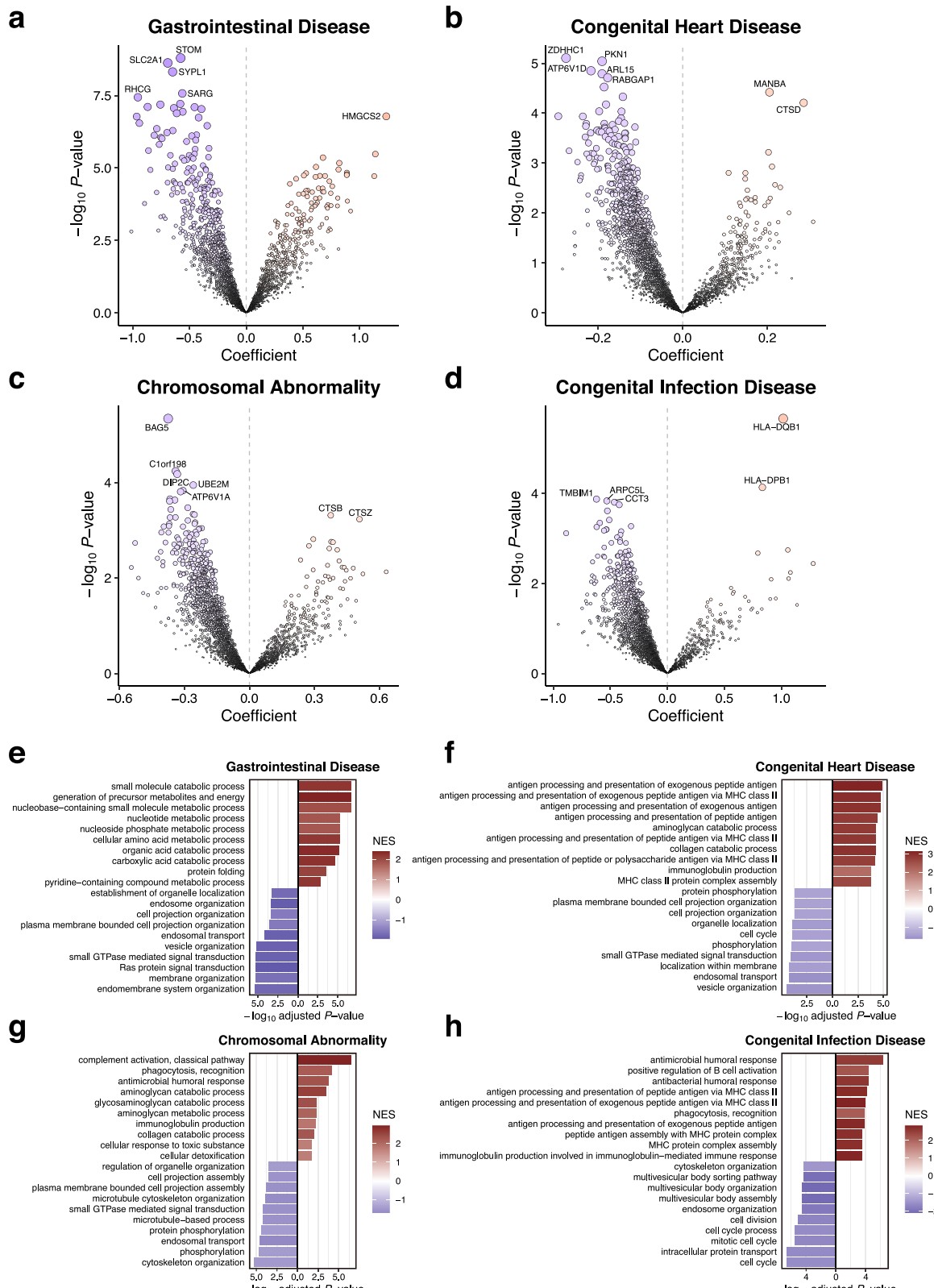

**Fig. 4 | Exploring meconium protein profiles for antenatal diseases.** Volcano plots showing the changes in the meconium proteome with GID (**a**), CHD (**b**), CA (**c**), and CID (**d**). The x-axis represents the effect size, indicated by the coefficient, while the y-axis displays the statistical significance, represented by -log₁₀(*P*-value). Positive coefficients indicate higher abundance in the disease samples, while negative coefficients indicate lower abundance. *P*-values for each coefficient were calculated using a two-sided *t*-test without adjustment for multiple testing. Bar plots showing the top 10 significantly enriched Gene ontology (GO) terms of proteins that are more abundant in the disease samples (red; GID (**e**), CHD (**f**), CA (**g**), and CID (**h**)) and less abundant in the disease samples (blue). *P*-values were calculated using a permutation test with Benjamini–Hochberg correction, based on the GSEA algorithm as implemented in the R package fgsea. NES normalised enrichment scores.

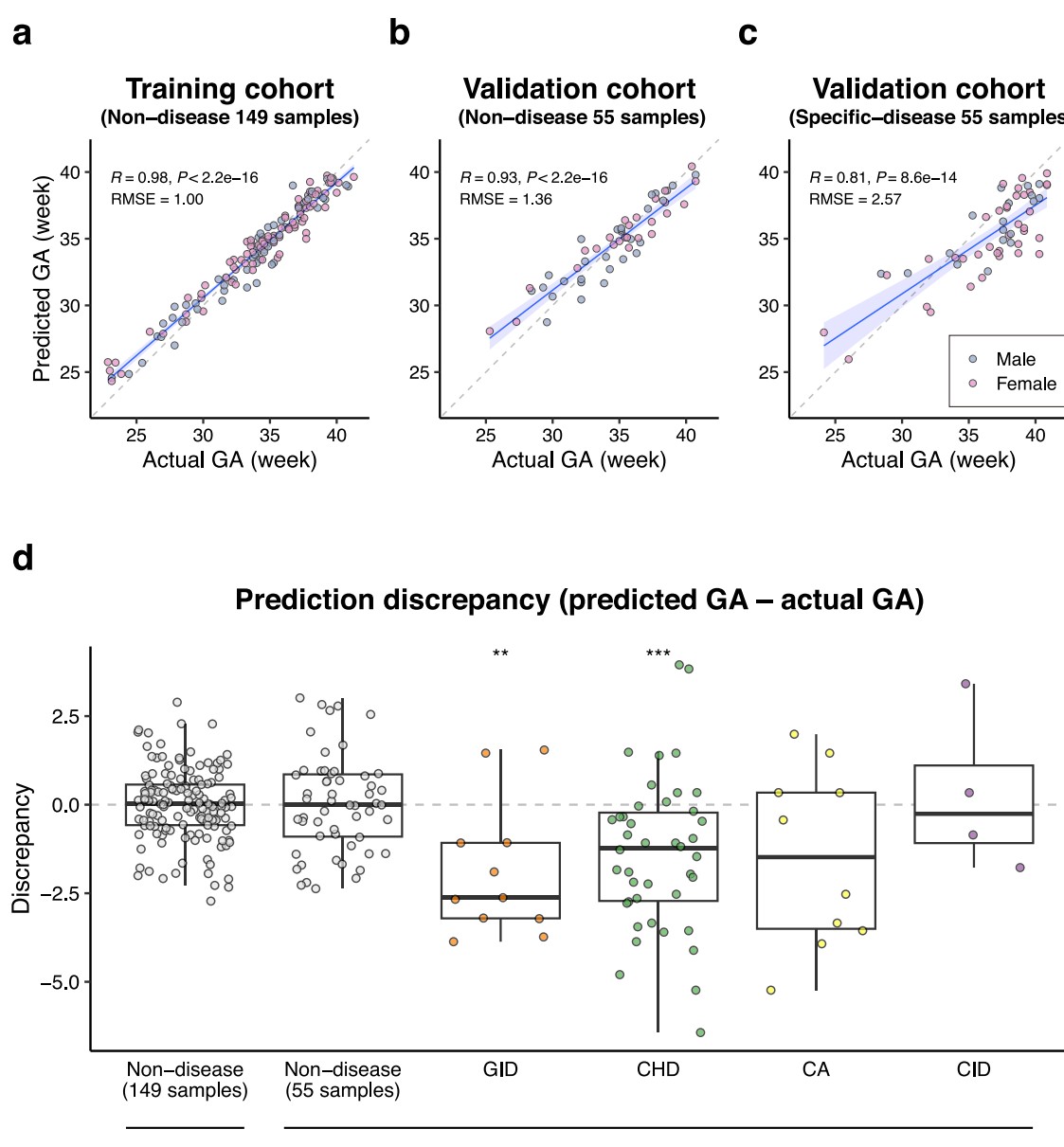

**Fig. 5 | Potential for prediction of gestational age by meconium proteomic analysis. a** Gestational age (GA) prediction in the training cohort (excluding specific diseases, 149 samples). RMSE root-mean-square error. Shaded areas around regression lines represent the 95% confidence interval. **b** GA prediction in the validation cohort (excluding specific diseases, 55 samples). RMSE root-mean-square error. Shaded areas around regression lines represent the 95% confidence interval. **c** GA prediction in the validation cohort (specific diseases, 55 samples). The blue and red dots represent males and females, respectively. Pearson correlation coefficients and two-sided P-values between the actual GA and predicted GA are shown. RMSE: root-mean-square error. Shaded areas around regression lines represent the 95% confidence interval. **d** Comparison of the difference between the actual GA and predicted GA among the cohorts: Training Non-disease (149 samples) vs. Validation Non-disease (55 samples) vs. Specific diseases (GID: 11 samples, CHD: 42 samples, CA: 10 samples, CI: 4 samples). GID gastrointestinal disease, CHD congenital heart disease, CA chromosomal abnormality, CID congenital infection disease. Statistical analyses compared with the Validation Non-disease cohort were performed using the Wilcoxon rank-sum test; **$P = 0.002$, ***$P = 0.0002$. Centrelines within box plots represent medians. Box limits indicate 25th and 75th percentiles, and the whiskers extend to 1.5 times IQR of 25th and 75th percentiles.

samples. Furthermore, as demonstrated in our study, these uncertainties can be addressed through validation, using methods like ELISA and other target-based high throughput approaches, such as selected reaction monitoring. The establishment of an ELISA system for quantifying meconium proteins, such as Mucin-2, identified in this study could provide a novel, non-invasive testing method for neonates within a clinical context.

In conclusion, we described robust human-derived protein profiles in the first meconium across various gestational ages using deep proteomic analyses. The analysis of meconium focusing on host-derived proteins holds significant promise not only for elucidating the gastrointestinal physiology of newborns but also for shedding light on the pathophysiology of systemic diseases such as GID, CHD, CA, and CID, as well as to guide clinical care strategies. A detailed investigation of host-derived proteins in meconium may lead not only to the discovery of new biomarkers and the elucidation of unknown aetiologies of neonatal diseases but also to the establishment of a novel method for evaluating the gastrointestinal systems of newborns. Further large-scale studies of meconium proteins are warranted to establish the validity and utility of our findings.

## Methods

### Characteristics of the human cohort

To assess the protein composition of meconium, we analysed samples collected from 259 newborns (Table 1 and Table 2). This prospective study involved the recruitment of newborns receiving care at the University of Tokyo Hospital, as illustrated in Fig. 1b. Newborns for whom consent from their parents could not be obtained, those from whom initial meconium could not be collected, or those born outside the hospital were excluded from the study. In this study, we exclusively gathered data regarding participants' sex, which is a biological attribute, and consistently referred to it by its accurate terminology. The sex variable was categorized into two groups: Male and Female. Overall, our study encompassed 112 male participants and 147 female participants, with a male-to-female ratio of 43.2% to 56.8%. To assess the external validation cohort, we analysed samples collected from 79 non-diseased newborns (GA: $34.1 \pm 4.3$ weeks). This study encompassed 40 male participants and 39 female participants, with a male-to-female ratio of 50.6% to 49.4%. Additionally, 22 males and 18 females, with a male-to-female ratio of 55.0% to 45.0% contributed another 40 samples of mucin 2 for ELISA analysis (GA: $31.9 \pm 5.5$ weeks). Notably, we did not collect information on gender identity in all three cohorts because the subjects were early postnatal newborns.

### Diagnostic methods for various diseases

In our study, samples associated with various diseases such as GID, CHD, CA, and CID were included. (Supplementary Data 4 and Supplementary Fig. 4a). Various diagnostic methods were employed for different diseases. GID was diagnosed using abdominal X-rays, ultrasound (US), and operative findings. CHD was primarily identified through postnatal US, with the majority of cases being detected during foetal ultrasonographic examination. Some cases underwent further evaluation with contrast-enhanced CT scans once their condition had stabilized. CA was diagnosed using techniques such as G banding or fluorescent in situ hybridization (FISH). In the case of CID, cytomegalovirus (CMV) infections were identified through the PCR analysis of urine samples, while bacterial infections were confirmed through culture findings from nasal cavity, stool, blood, and skin samples. These diagnostic procedures were complemented by X-ray imaging, blood tests (including inflammatory markers like white blood cell count and C-reactive protein levels), and clinical assessments.

### Estimating the anticipated delivery date

Determining the anticipated delivery date involves employing various methods outlined in the "Guideline for Obstetrical Practice in Japan 2023." These methods consider factors such as the embryo transfer date and identifiable ovulation date, normal menstrual cycle and last menstrual period, or crown-rump length (CRL) at specific stages of pregnancy. Typically, the calculation involves measuring the biparietal diameter (BPD) after one week, with the expected delivery date typically pinpointed around the 13th week and 6th day of pregnancy. In the Japanese obstetric care scenario, pregnant individuals promptly seek consultation at obstetric and gynaecology departments upon confirming pregnancy. Therefore, GA determination often relies on CRL measurements rather than solely on estimates derived from the last menstrual period. This paper adopts an optimized approach integrating both the last menstrual period and US measurements to accurately establish GA.

### Meconium proteome measurements

The meconium samples were collected from 338 individuals, of which 259 and 79 were for screening and validation studies, respectively; thus, the proteome analysis was performed with 338 biological replicates.

Each meconium sample was stored at $-20\,^\circ$C to preserve its integrity. Proteomic analysis of the meconium samples was performed according to a previously established protocol[19]. Briefly, soluble proteins from the meconium were extracted in Tris-buffered saline containing Tween 20 and protease inhibitors (CAT# 5892791001, complete ULTRA Tablets, Sigma-Aldrich). The extraction process involved pipetting and inversion after incubating the samples on ice for 30 min. Subsequently, the samples were centrifuged at $15,000 \times g$ for 15 min at $4\,^\circ$C, and supernatants were collected. The extracted proteins were quantified using a Pierce™ BCA Protein Assay Kit (Thermo Fisher Scientific, Waltham, MA, USA) at 300 ng/µL. Reductive alkylation was performed on the collected samples, followed by cleaning using the SP3 method. Enzymatic digestion with trypsin/Lys-C (Promega, Madison, WI, USA) was conducted at $37\,^\circ$C overnight. The digestates were purified using GL-Tip SDB (GL Sciences, Tokyo, Japan) according to the manufacturer's protocol. The peptides were dissolved again in 2% acetonitrile (ACN) containing 0.1% trifluoroacetic acid and quantified using a BCA assay at 200 ng/µL. Following pretreatment for shotgun proteome analysis, the peptides (400 ng) were directly injected onto an analytical column 15 cm × 75 µm, C18, 1.6 µm (Aurora Column, Ion Opticks, Australia) and then separated with a 90 min gradient (mobile phase A = 0.1% formic acid (FA) in water, B = 0.1% FA in 80% ACN) consisting of 0 min 6% B, 78 min 36% B, 84 min 65% B, 90 min 65% B at a flow rate of 150 nL/min using an UltiMate 3000 RSLCnano LC system (Thermo Fisher Scientific). The peptides eluted from the column were subjected to analysis using the Orbitrap Exploris 480 (Thermo Fisher Scientific) following a DIA approach with overlapping windows[61]. The spray voltage and capillary temperature for mass spectrometry (MS) were set at 1.5 kV in the positive ion mode and 275 °C, respectively. The MS1 scan range was set as full scan with m/z 495–785 at a mass resolution of 15,000 to set an auto gain control (AGC) target of $3 \times 10^6$ and maximum injection time of 'Auto'. The MS2 were collected at m/z 200–1800 at 30,000 resolution to set an AGC target of $3 \times 10^6$, maximum injection time of 60 ms, and stepped normalised collision energy of 22, 26, and 30%. An isolation width for MS2 was set to 4 Th, and overlapping window patterns in 500–780 m/z were employed for window placements optimized using Xcalibur 4.3 (Thermo Fisher Scientific).

The mass spectrometry files were searched against a human spectral library using DIA-NN v1.8[62]. A human spectral library was generated from the human protein sequence database (UniProt id UP000005640, reviewed, canonical, downloaded on 31 March, 2021) using DIA-NN. The following parameters were used to create the spectral library: digestion enzyme, trypsin; missed cleavages, 1; peptide length range, 7–45; precursor charge range, 2–4; precursor m/z range, 490–790; fragment ion m/z range, 200–1800; and "FASTA digest for library-free search/library generation," "deep learning-based spectra, RTs, and IMs prediction," "n-term M excision," and "C carbamidomethylation." The DIA-NN search parameters were as follows: mass accuracy, 10 ppm; MS1 accuracy, 10 ppm; protein inference, genes; neural network classifies, single-pass mode; quantification strategy, robust LC (high precision); cross-run normalisation, RT-dependent; and "unrelated runs," "use isotopologues," "MBR," "heuristic protein inference," and "no shared spectra" were enabled. The protein identification threshold was set at 1% or less for both the precursor and protein FDRs. Proteins with at least one unique peptide were selected.

### Association of phenotypes with protein expression levels

A total of 3433 proteins detected in at least 100 samples were used for subsequent analyses. The protein expression levels were log 10-transformed. Missing values for undetected proteins were imputed based on a normal distribution with a median shifted from the median of the measured data distribution of the detected proteins towards low expression (down-shift of 1.8 and width of 0.3)[63]. A list of tissue-specific proteins was obtained from the 'total elevated' gene list in the Human Protein Atlas[64] (https://www.proteinatlas.org/humanproteome/tissue/

tissue+specific; downloaded on February 28, 2024). GO term over-representation analysis of proteins was performed using the enrichGO function of the R package clusterProfiler (v4.6.2)[65]. *P*-values were adjusted using the Benjamini−Hochberg method.

To evaluate the association between GA, sex, and protein expression levels, the following linear model was used:

$$\text{Protein expression level} \sim \alpha + \beta 1 \text{ gestational age} + \beta 2 \text{ sex} + \varepsilon.$$

Similarly, the following linear model was fitted to evaluate the effects of specific diseases on protein expression levels:

$$\text{Protein expression level} \sim \alpha + \beta 1 \text{ gestational age} + \beta 2 \text{ sex} + \beta 3 \text{ disease} + \varepsilon.$$

The coefficients (effect sizes) and associated *P*-values for GA, sex, and disease were extracted from each fitted model. Volcano plots were visualised using the R package ggplot2 (v3.4.3). GSEA of the GO terms was performed using the gseGO function of the R package cluster-Profiler, where genes (proteins) were ranked based on their *P*-values and coefficients. GSEA of tissue-specific expression was performed using the R package fgsea (v1.24.0)[66], where the list of tissue-specific proteins was used as the gene set.

### Clustering of protein trajectories during gestational ageing

Protein expression levels were z-scored, and LOESS regression was fitted for each protein during gestational ageing. Hierarchical clustering analysis was performed using complete linkage and Euclidean distance to group proteins with similar trajectories.

### GA prediction using proteome data

The dataset was split into training and validation cohorts as described above. The LASSO regression method, implemented in the R package glmnet (v4.1.8)[67], was employed to model the relationship between GA and meconium proteome data. For training the model, 149 non-disease-associated samples were used. A cross-validation approach was used to determine the optimal regularization parameter lambda. The GA was predicted for the training and validation cohorts using the trained LASSO model.

### Measuring meconium MUC2 concentrations

Meconium samples were initially diluted to a concentration of 10 μg per 60 μL in TBS, with the addition of protease inhibitor cocktail (Roche). After pipetting, they were kept on ice for 30 min, and subsequently centrifuged at 15,000 *g* for 30 min at 4 °C. The resulting supernatant was used for ELISA. Supplemental Fig. 3 was conducted as a singleton. To quantify the MUC2 concentration, a Human MUC2 ELISA Kit (Cloud Clone Corp) was used, following the manufacturer's instructions. The samples were applied to a pre-coated plate containing a biotin-conjugated antibody specific for MUC2 and incubated for 1 h. After thorough washing, avidin conjugated to HRP was added to the plate for 30 min, followed by another round of washing and TMB reaction. The enzyme-substrate reaction was terminated by adding sulphuric acid solution, and the resulting colour change was measured spectrophotometrically at a wavelength of 450 nm (PerkinElmer). The concentration of MUC2 in the samples was determined by comparing the optical density (OD) of the samples with a standard curve. Sample dilutions were adjusted to align with the standard curves.

### Reporting summary

Further information on research design is available in the Nature Portfolio Reporting Summary linked to this article.

## Data availability

The data supporting the findings from this study are available within the manuscript and its supplementary information. The MS data of 259 samples and 79 external samples used in this study are available in the ProteomeXchange Consortium via the jPOST partner repository[68] under the accession codes PXD047426 for ProteomeXchange and JPST002405 for jPOST, PXD050164 for ProteomeXchange, and JPST002961 for jPOST, respectively. Any additional raw data will be available from the corresponding authors upon reasonable request. Source data are provided with this paper.

## Code availability

The code used in this paper can be obtained from GitHub (https://github.com/my0916/meconium).

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

## Acknowledgements

We extend our gratitude to the nurses and attending physicians who were instrumental in addressing meconium-related issues and who provided an overview of this research to the parents of the study cohorts. We express our sincere appreciation to the parents for their invaluable participation in this study. E.W. is funded through a Grant-in-Aid for Scientific Research from the Japan Society for the Promotion of Sciences (JSPS) (no. 19K24007 and no. 23K15465), the Kawano Masanori Public Interest Incorporated Foundation for Promotion of Pediatrics, and the Gunma Foundation for Medicine and Health Science. J.F. received funding from a Grant-in-Aid for Scientific Research from the JSPS (no. 20K20469).

## Author contributions

Y.S. and E.W. contributed to the overall study design. Y.S., K.K., A.I., T.M., G.K., S.K., and E.W. conducted the clinical analyses. M.Y., R.K., M.I., E.K., and Y.K. performed proteome analysis. Y.S., K.K., A.I., T.M., T.K., and E.W. were responsible for the clinical interpretation. The initial draft was prepared by Y.S. and E.W. with input from the authors who provided valuable editorial comments. N.T., J.F., and O.O. supervised the study. Y.K. and E.W. serve as the corresponding authors of this study.

## Competing interests

The authors declare no competing interests.

## Ethics

Ethical approval was granted by the Institutional Review Board of the University of Tokyo Hospital (number 2019010NI-13). Additionally, informed consent was obtained from the parents or legal guardians of all participants. All methods were conducted in strict accordance with the ethical guidelines outlined in the Japanese Ethical Guidelines for Medical and Biological Research Involving Human Subjects.
