## [Peer Review File · Nature Communications]

REVIEWER COMMENTS

Reviewer #1 (Remarks to the Author):

This is an interesting manuscript describing the protein composition of meconium at different gestational ages and focuses on mucin.

Here are some itemized comments:

- The title could emphasize the range of gestational ages because the authors have studied meconium, not only at term but in preterm neonates as well. This makes the article of greater interest.
- The abstract should specify the time for collection of meconium. It is one thing to collect samples immediately after birth or after feeding has occurred.
- One of the objectives of the study is gestational age prediction based on meconium composition. The authors need to compare this with the gestational age derived from the combination of the last menstrual period and fetal biometry, and also with the results of neonatal examination.
- There needs to be some more information about the indications for preterm delivery. Some preterm deliveries occur after the spontaneous onset of labor either with intact or ruptured membranes. The prevalence of infection and inflammation is different in these groups compared to patients who have fetal growth restriction or preeclampsia. It is important to know what the indications were for delivery.
- It is quite interesting that meconium is different in males than in females. This is illustrated in Figure 2 and supplementary table 1.
- The statement that the fetus does not pass meconium has been challenged (see Gallo et al. in the American Journal of Obstetrics & Gynecology 2023 and also Carlos Lopez Ramon y Cajal).
- It may be better to refer to preterm gestations rather than short gestation neonates as this is not used in clinical medicine. We refer to preterm gestations.
- Please confirm that the method of proteomic analysis, which is mass spectrometry based, allows detection of the number of proteins described by the authors.
- The gold standard for gestational age needs to be explained. Specifically, how was GA ascertained?
- The authors present the differences between neonates with gastrointestinal disease, congenital heart disease, chromosomal abnormalities, and congenital infection in Figure 5. More information is required about these clinical conditions. For example, congenital infections could be viral or

bacterial. The latter would refer to early onset neonatal sepsis. Is this the condition that the authors are describing?

- If meconium is able to identify neonates with early onset sepsis, this is a major finding and requires more description than what is provided in the material and methods section. It will necessary to know how the diagnoses were made because the terms gastrointestinal disease, congenital heart disease, and chromosomal abnormalities are very broad.
- There are other questions. Is the meconium of neonates born to mothers with gestational diabetes different from those without complications of pregnancy? There are 23 cases with gestational diabetes. The same can be said about hypertensive disorders of pregnancy when there are 22 cases and also with funisitis, which is a manifestation of the fetal inflammatory response syndrome (see Jung et al for more details).
- I believe that this is a unique study and an important contribution that could be highly cited. 10% of pregnancies are complicated by meconium stained amniotic fluid and if meconium can be used as a source of biomarkers, it offers a non-invasive method to assess fetal status at birth. Meconium has been used to screen for maternal ingestion of drugs and this is a precedent for the use of meconium for other purposes. For example, it would be very useful if meconium could be used to assess the likelihood of fetal inflammation diagnosed as the presence of funisitis.

Reviewer #2 (Remarks to the Author):

Thank you for the opportunity to review Shitara et al.'s manuscript, "Host-derived protein profiles of meconium from human neonate", with a specific focus on the analytical aspects of the study.

The authors report the results of a high-content proteomic analysis of meconium samples from 259 neonates using a Data Independent Acquisition (DIA) proteomics approach. Up to 5,370 proteins are identified per sample, with the majority of samples containing ~3,000 proteins. Several analyses are performed. First, the authors build a linear model with individual proteins as response variables and sex and gestational (GA) age as independent variables. This analysis allows identification of differentially expressed proteins between males and females, as well as the association between individual proteins and GA. Second, a multivariable approach using Lasso is performed to identify a predictive model of GA based on the analysis of the proteomic dataset. For this analysis, samples are divided into training and validation sets, and excluding samples from diseased neonates is necessary to train the Lasso model. The analysis identifies a robust predictive model of GA containing 64 proteomic features with non-zero coefficients ($R=0.98$ training, $R=0.83$ validation). The validation is performed on samples from both diseased and healthy neonates.

Overall, the study has many strengths, including the DIA approach for the analysis of clinically relevant meconium samples, a focus on clinically important variables (sex and GA), the appropriate use of the Lasso method given the large feature dataset relative to the sample size, and inclusion of a training and validation sample set. However, the analyses suffer from several weaknesses. Addressing the following comments would strengthen the manuscript:

1. The interaction between GA and disease status complicates the predictive model analysis of GA. As such, the authors revert to a two-step analysis excluding disease samples from the training set after a first analysis on the entire dataset was performed. Instead, it would be preferable to perform two separate Lasso analyses, each one with a clearly defined goal: 1) prediction of GA in healthy neonates, 2) classification of healthy vs. disease samples.
2. When reporting Lasso models, performance metrics such as RMSE would be useful.
3. The performance of the Lasso model on the training set that includes samples from diseased neonates should be reported.
4. In the methods describing the Lasso analysis, please specify the input feature dataset (i.e., the number of proteins included).
5. Given that the number of proteins identified varies between samples, how is missing data handled by the analysis?
6. The results of the GA analysis shown in Fig. 3 (simple linear model) are somewhat redundant with the Lasso analysis in Fig. 4. Please provide a justification for performing two different analyses of GA.
7. Similarly, a rationale for using proteins as dependent variables for the analysis of sex differences should be provided. A Lasso method could have also been utilized to classify samples by sex."

Reviewer #3 (Remarks to the Author):

The manuscript entitled "Host-derived protein profiles of meconium from human neonates" conducted a deep proteomic analysis of meconium and established a gestational age prediction model. While we appreciate the effort put into this study, we have identified several concerns that limit the novelty and depth of the research.

1. There have been previous studies focusing on the proteomic analysis of meconium proteins and conducted pathway enrichment (also enriched proteins associated with the extracellular matrix) (Lisowska-Myjak et al., *J Obstet Gynaecol Res.* 2019; 45:556-64), which may limit the innovation of the study.

2. The results presented in this paper are potentially of interest to the readers. However, they are based on a very limited number of experimental data points, without in-depth exploration of the molecular mechanisms and translational results of the findings, which limited the confidence in the claimed results.

3. The authors grouped a total of 259 samples, conducting screening for biomarkers in a subset and subsequent validation in another subset. Despite the random grouping, the samples were sourced from the same institution at the exact location and collected within a similar timeframe. This approach of screening and validating within a constrained setting limits the generalizability of the model, and its scientific validity requires further verification.

4. Additionally, the authors identified 55 samples (approximately 22% of the total) that did not conform to the model and were subsequently excluded. A more rigorous scientific justification is needed for the exclusion of these samples, along with subsequent arguments regarding the applicability of the model.

5. As mentioned above, the exclusion of 55 samples may suggest that the protein composition of meconium is still largely influenced by amniotic fluid proteins. It is worth noting that amniotic fluid can be obtained non-invasively (similar to meconium), and the author may need to discuss the advantages and disadvantages of using amniotic fluid for protein analysis compared to meconium. Additionally, consideration could be given to integrating the analysis of both non-invasive sample types to broaden the applicability of the model. This discussion could encompass the potential impact of amniotic fluid proteins on meconium protein composition.

6. The author ultimately employed a linear fitting approach to associate variables, including protein expression levels, gestational age, and sex. Further elaboration is needed to justify the choice of a relatively simple linear model over more complex models and discuss which model offers a better fit. Additionally, the impact of log transformation on the predictive performance of the model when dealing with protein expression levels should be addressed, and a detailed explanation of the data standardization process is required.

7. The manuscript declared that early ultrasound detection is the most accurate method for assessing gestational age. However, the methods described in the text require non-invasive collection after the birth of the fetus, which may introduce a temporal delay for gestational age assessment. The authors need to engage in further discussion on this aspect.

8. The results in Figure 5 indicate that the 64 selected biomarkers exhibit different expression patterns in the four diseases. Based on the enrichment results, can the authors explain the possible mechanisms underlying the occurrence of the four different diseases. This could help elucidate the reliability of the predictive model.

9. Figure 5e-h, the author found significant changes in metabolism-related proteins, which can corroborate previous reports (Petersen et al., *Cell Rep Med.* 2021;2:100260). Could the author

discuss the findings based on the metabolome of meconium and elaborate on the advantages of proteomic studies and how they complement existing research results.

10.The arrangement and content of the figures may not meet the requirements of this journal.

We express our sincere gratitude to the reviewers for identifying areas of improvement in our
manuscript submitted to *Nature Communications*. We have diligently worked to address each query
raised and have made corresponding revisions to the manuscript. We trust that our responses
adequately address the raised concerns.

**Reviewer #1:**

**Specific comment 1:** *The title could emphasize the range of gestational ages because the authors*
*have studied meconium, not only at term but in preterm neonates as well. This makes the article of*
*greater interest.*

**Response:** We express our gratitude to the reviewer for the insightful suggestion regarding the title.
Below, we present the revised title.

“Host-derived protein profiles of human neonatal meconium across gestational ages”

**Specific comment 2:** *The abstract should specify the time for collection of meconium. It is one thing*
*to collect samples immediately after birth or after feeding has occurred.*

**Response:** We express our gratitude to the reviewer for the insightful suggestion regarding the
abstract. The majority (76.4%) of the first meconium samples used in our study were collected
before feeding, whereas a portion (23.6%) were obtained after feeding. The median and mean times
for the first meconium passage from the anus were 9.1 h (IQR: 4.5–14.4 h) and 11.9 ± 9.47 h,
respectively. The collection times ranged from 0 to 69.5 h. This description has been incorporated
into the Results section (**page 6, lines 97–100**).

In the abstract, we added the following sentences “The first meconium samples were collected, with
the majority obtained before feeding, and the mean time for the first meconium passage from the
anus was 11.9 ± 9.47 h” (**page 3, lines 41–43**).

**Specific comment 3:** *One of the objectives of the study is gestational age prediction based on*
*meconium composition. The authors need to compare this with the gestational age derived from the*
*combination of the last menstrual period and fetal biometry, and also with the results of neonatal*
*examination.*

**Response:** We express our gratitude to the reviewer for their invaluable suggestions. In the obstetric
care landscape of Japan, pregnant women promptly seek consultation at an obstetrics and
gynaecology department upon confirming pregnancy. Subsequently, at 8 to 10 weeks of gestation,
precise measurements of the Crown Rump Length (CRL) are conducted to ascertain the accurate
expected delivery date. Consequently, in the cases considered in this study, gestational age was
determined based on the CRL rather than relying solely on estimates derived from the last menstrual

period. To elaborate, the gestational age in this paper was established using an optimized approach
that integrates both the last menstrual period and ultrasound measurements, aligning with the
methodology proposed by the reviewer. Thus, we assert that any potential inaccuracies in gestational
age determination do not impact the observed relationship between gestational age and meconium
composition presented in this study. The methodology used for estimating the gestational age is
described in detail in the Method section under “Estimating the anticipated delivery date” (**page 19,**
**line 331**).

**Specific comment 4:** *There needs to be some more information about the indications for preterm*
*delivery. Some preterm deliveries occur after the spontaneous onset of labor either with intact or*
*ruptured membranes. The prevalence of infection and inflammation is different in these groups*
*compared to patients who have fetal growth restriction or preeclampsia. It is important to know*
*what the indications were for delivery.*

**Response:** We extend our appreciation to the reviewer for the valuable suggestion regarding the
indications for preterm delivery.

Of the 259 samples, 163 were preterm deliveries, accounting for 62.9% of the total. The leading
clinical condition contributing to preterm delivery was threatened premature birth, with 57 samples,
followed by non-reassuring foetal status with 38 samples. We have included a new supplementary
table specifically addressing these findings (**Supplementary Table 2**). Additionally, **Table 1**
presents data on the premature rupture of membranes (**Table 1**).

**Specific comment 5:** *It is quite interesting that meconium is different in males than in females. This*
*is illustrated in Figure 2 and supplementary table 1.*

**Response:** We greatly appreciate your valuable feedback. Your insightful comments served as
encouragement for us to revise our paper. We adjusted **Fig. 2** according to the revised paper and
added **supplementary Table 3**.

**Specific comment 6:** *The statement that the fetus does not pass meconium has been challenged (see*
*Gallo et al. in the American Journal of Obstetrics & Gynecology 2023 and also Carlos Lopez*
*Ramon y Cajal).*

**Response:** We sincerely appreciate the invaluable information provided by the reviewer. We were
unaware of this paper that comprehensively documents new findings concerning meconium in the
fetus. Consequently, we have referenced this seminal paper in our revised manuscript (**page 5, lines**

77–78). In addition, we added future insights regarding the study of meconium using amniotic fluids
(page 16, lines 275–278).

**Specific comment 7:** *It may be better to refer to preterm gestations rather than short gestation*
*neonates as this is not used in clinical medicine. We refer to preterm gestations.*

**Response:** We extend our appreciation to the reviewer for the recommended terminology adjustment
related to gestation. Consequently, we have revised “short gestation” to “preterm gestation.”

**Specific comment 8:** *Please confirm that the method of proteomic analysis, which is mass*
*spectrometry based, allows detection of the number of proteins described by the authors.*

**Response:** We express our gratitude to the reviewer for the insightful suggestion regarding the
number of proteins detectable using proteome analysis. In recent years, mass spectrometry has
undergone significant advancements, enabling the detection of over 10,000 proteins within a single
run in cellular or tissue samples (Kawashima et al., *Int J Mol Sci*, 2019). In our laboratory, utilizing
this method, we have successfully identified approximately 9,500 proteins in HEK293 digestive
samples, and over 2,000 proteins in stool proteome analyses, as previously reported (Watanabe et al.,
*proteomes*, 2019). Therefore, using the method described in our paper, the detection of more than
5,000 proteins in the meconium samples is not unexpected.

We added the following explanation in the Results section: “In recent years, mass spectrometry has
undergone significant advancements, enabling the detection of over 10,000 proteins within a single
run in cellular or tissue samples^{16–18}. Utilizing this method, we previously identified over 2,000
proteins in stool proteome analysis¹⁹” (page 4, lines 66–68).

**Specific comment 9:** *The gold standard for gestational age needs to be explained. Specifically, how*
*was GA ascertained?*

**Response:** We express their gratitude to the reviewer for their invaluable suggestions. Most pregnant
women undergoing health check-ups in Japan, including those in this study, have their gestational
age determined using an optimized method that combines data from the last menstrual period and
crown–rump length (CRL) measurement in early pregnancy. According to the “Obstetrics and
Gynecology Clinical Practice Guidelines 2023” method, the expected delivery date is calculated by
considering either 1) the embryo transfer date and identifiable ovulation date, 2) a normal menstrual
cycle and last menstrual period, or 3) the CRL at 8–10 weeks of pregnancy or 11 days of pregnancy.
This calculation is performed using the major transverse diameter (BPD) after 1 week, and the
expected delivery date is determined by the 13th week and 6th day of pregnancy. This additional

information has been incorporated in the Methods section (**page 19, line 331**) (please refer to our
response to Specific comment 3).

**Specific comment 10:** *The authors present the differences between neonates with gastrointestinal*
*disease, congenital heart disease, chromosomal abnormalities, and congenital infection in Figure 5.*
*More information is required about these clinical conditions. For example, congenital infections*
*could be viral or bacterial. The latter would refer to early onset neonatal sepsis. Is this the condition*
*that the authors are describing?*

**Response:** We express our gratitude to the reviewer for this comment. In the revised manuscript, the
specific disease proteome is described in the Results section under “Disease-dependent meconium
protein profiles” (**page 11, line 197; Fig. 4**). As demonstrated in **Supplementary Table 8**, there
were three cases of bacterial infection and one case of viral infection, with no instances of
septicemia observed. Details regarding gastrointestinal diseases, congenital heart diseases, and
chromosomal abnormalities are also provided in **Supplementary Table 8**.

**Specific comment 11:** *If meconium is able to identify neonates with early onset sepsis, this is a*
*major finding and requires more description than what is provided in the material and methods*
*section. It will necessary to know how the diagnoses were made because the terms gastrointestinal*
*disease, congenital heart disease, and chromosomal abnormalities are very broad.*

**Response:** Thank you for your feedback. We had anticipated discovering specific biomarkers for
early-onset sepsis in the neonatal period. However, our preliminary study had a limited number of
cohorts, and no clinical septic cases were observed. We are now planning multicentre cohort studies
focused on these challenging clinical cases. Additionally, we have provided detailed information on
gastrointestinal disease (GID), congenital heart disease (CHD), chromosomal abnormality (CA), and
congenital infection disease (CID) in **Supplementary Table 8**. As described in “Diagnostic methods
for various disease” in the Methods section (**page 18, line 318**), GID was diagnosed through
abdominal X-rays, ultrasound (US), and operative findings. CHD was primarily diagnosed via
postnatal US, with most cases detected during foetal ultrasonographic examination. Some cases were
further investigated with contrast-enhanced CT scans once stable. CA was diagnosed using G
banding or fluorescent in situ hybridization (FISH). For CID, cytomegalovirus (CMV) infection was
diagnosed using PCR analysis of urine samples, while bacterial infections were diagnosed through
culture findings from nasal cavity, stool, blood, and skin samples, complemented by X-ray imaging,
blood tests (including inflammatory markers such as white blood cell count and C-reactive protein
levels), and clinical assessments. In the case of CAM, the sample size was limited, and there were
missing values in the proteome analysis, preventing detailed investigation in this preliminary paper

(page 15, lines 268–269). However, as demonstrated in the GSEA analysis, CID displayed a unique
protein profile, suggesting that the foetus was influenced by the intrauterine environment originating
from the mother (page 12, line 213 to page 13, line 218). Nonetheless, as the reviewer rightly
pointed out, it is crucial to assess these details in future studies with well-established cohorts.

**Specific comment 12:** *There are other questions. Is the meconium of neonates born to mothers with*
*gestational diabetes different from those without complications of pregnancy? There are 23 cases*
*with gestational diabetes. The same can be said about hypertensive disorders of pregnancy when*
*there are 22 cases and also with funisitis, which is a manifestation of the fetal inflammatory*
*response syndrome (see Jung et al for more details).*

**Response:** We express our gratitude to the reviewer for their invaluable comments. We have
significant interest in exploring the relationship between maternal factors and the composition of the
first meconium. Specifically, we conducted an analysis of the meconium proteome in cases of
gestational diabetes mellitus (GDM) and hypertensive disorders of pregnancy (HDP). Our findings
reveal enrichment of transport functions in babies born to mothers with GDM, whereas babies born
to mothers with HDP exhibit enrichment of metabolism functions (page 13, lines 218–222). These
results are illustrated in the newly added **Supplementary Fig. 4b and 4c**. At our hospital, placental
pathological analysis is typically overseen by gynaecologists. Of our cohort of 259 samples, 212
underwent pathological assessment, while 47 samples did not undergo investigation. Consequently,
conditions such as chorioamnionitis and funisitis were not analysed or included in this study due to
the missing values. The presence of missing values has been indicated in **Table 1**. We incorporated
this limitation into the Discussion section (page 15, lines 268–269).

**Specific comment 13:** *I believe that this is a unique study and an important contribution that could*
*be highly cited. 10% of pregnancies are complicated by meconium stained amniotic fluid and if*
*meconium can be used as a source of biomarkers, it offers a non-invasive method to assess fetal status*
*at birth. Meconium has been used to screen for maternal ingestion of drugs and this is a precedent for*
*the use of meconium for other purposes. For example, it would be very useful if meconium could be*
*used to assess the likelihood of fetal inflammation diagnosed as the presence of funisitis.*

**Response:** We sincerely value your feedback, as it provides valuable insight that motivates us to refine
our paper. We are delighted that you have expressed interest in our meconium study, particularly in
its non-invasive collection method for clinical samples. As described above (response to specific
comment #12), in our hospital, gynaecologists typically oversee placental pathological analysis. Of
the 259 samples in our cohort, 212 underwent pathological assessment, while 47 did not. Consequently,
conditions such as chorioamnionitis (CAM) and funisitis were not included in this study. We

incorporated this limitation into the Discussion section (**page 15, lines 268–269**). However, given the
potential of the meconium proteome to predict intrauterine infection, CAM, and funisitis, as the
reviewer suggested, it is crucial to explore protein biomarkers for these clinical issues in our future
studies. Therefore, we also included the following sentences in the Discussion section: “Establishing
a correlation between the proteomics of amniotic fluid and meconium could lay the groundwork for
future research. This revelation hints at the possibility of harnessing human meconium insights to
develop biomarkers through the amniocentesis approach” (**page 16, lines 275–278**).

**Reviewer #2:**

**Specific comment 1:** *The interaction between GA and disease status complicates the predictive model*
*analysis of GA. As such, the authors revert to a two-step analysis excluding disease samples from the*
*training set after a first analysis on the entire dataset was performed. Instead, it would be preferable*
*to perform two separate Lasso analyses, each one with a clearly defined goal: 1) prediction of GA in*
*healthy neonates, 2) classification of healthy vs. disease samples.*

**Response:** We appreciate the Reviewer’s helpful suggestion. As suggested, we developed a model to
predict gestational age using only neonates without specific diseases (n = 149) as a training cohort
(**Fig. 5a**). This enabled us to predict the gestational age in neonates without specific diseases (n = 55)
with high accuracy (R = 0.93, RMSE = 1.36, **Fig. 5b**). In contrast, the accuracy of gestational age
prediction in neonates with specific diseases (n = 55) was not as high (R = 0.81, RMSE = 2.57, **Fig.**
**5c**). The predicted gestational age in neonates with specific diseases was significantly shorter than that
in neonates without specific diseases (P < 0.01, **Fig. 5d**), indicating the immaturity of the
gastrointestinal system in diseased neonates.

**Specific comment 2:** *When reporting Lasso models, performance metrics such as RMSE would be*
*useful.*

**Response:** We thank the Reviewer for this suggestion. In the revised manuscript, we have added
RMSE when presenting the predictive analysis results (**Fig. 5a–c, Supplementary Fig. 5a and 5c**).

**Specific comment 3:** *The performance of the Lasso model on the training set that includes samples*
*from diseased neonates should be reported.*

**Response:** We thank the Reviewer for this suggestion. In addition to the predictive model mentioned
above (response to comment 1; **Fig. 5**), we randomly divided all 259 samples, including samples from
diseased neonates, into training and validation cohorts at a 2:1 ratio and developed a model to predict

gestational age (**Supplementary Fig. 5a**).

**Specific comment 4:** *In the methods describing the Lasso analysis, please specify the input feature*
*dataset (i.e., the number of proteins included).*

**Response:** We appreciate the Reviewer's helpful suggestion. In the original manuscript, we used all
5,370 identified proteins for the analyses. However, we realized that some of these proteins were
detected in a limited number of samples. Therefore, in the revised manuscript we decided to use only
the 3,433 proteins that were detected in more than 100 samples. These 3,433 proteins were used as the
input for the Lasso analysis. In the Methods section of the revised manuscript, we have added the
following text:

"A total of 3,433 proteins detected in at least 100 samples were used for subsequent analyses. The
protein expression levels were log 10-transformed. Missing values for undetected proteins were
imputed based on a normal distribution with a median shifted from the median of the measured data
distribution of the detected proteins towards low expression (down-shift of 1.8 and width of 0.3)"
(**page 21, lines 366–369**).

**Specific comment 5:** *Given that the number of proteins identified varies between samples, how is*
*missing data handled by the analysis?*

**Response:** We thank the Reviewer for pointing this out. In the original manuscript, the expression
levels of undetected proteins were set to zero, which led to a significant difference compared to the
expression levels of detected proteins (**Figure R1a**). In the revised manuscript, we have imputed the
missing values for the undetected proteins based on a normal distribution with a median shifted from
the median of the measured data distribution of the detected proteins towards low expression. As a
result, the expression levels of all the proteins are now close to a normal distribution (**Figure R1b**).
This missing value imputation method is well-established in proteomics research (Tyanova *et al.*, *Nat*
*Methods*, 2016), as evidenced by its use in several studies (Ishikawa *et al.*, *J Proteome Res*, 2022; Sato
*et al.*, *ACS Omega*, 2022; Nakajima *et al.*, *Int J Mol Sci*, 2024). It is noteworthy that the overall results
did not change drastically even after imputing the missing values.

**Figure R1. Distribution of protein expression levels before (a) and after (b) missing value**
 **imputation.** Undetected proteins are shown in red and detected proteins are shown in green.
 Expression levels are log 10 transformed.

**Specific comment 6:** *The results of the GA analysis shown in Fig. 3 (simple linear model) are*
 *somewhat redundant with the Lasso analysis in Fig. 4. Please provide a justification for performing*
 *two different analyses of GA.*

**Response:** We thank the Reviewer for pointing this out. The purpose of the linear and non-linear
 trajectory analysis (**Fig. 3**) is to explore the proteomic changes with gestational age and to identify
 specific proteins that are significantly associated with gestational age, which could provide insights
 into the biological processes that may be influenced by gestational age. In contrast, the purpose of the
 predictive analysis (**Fig. 5**) is to evaluate the feasibility of predicting the gestational age based on the
 proteomic profile, which could also be used to predict the disease status. Indeed, some proteins used
 in the prediction model overlapped with the proteins that changed significantly with gestational age,
 but conducting both analyses provides us with a more comprehensive understanding of the relationship
 between gestational age and the proteome. In the revised manuscript, we clarified the purposes of these
 two different analyses (**page 9, lines 153–156; page 13, lines 227–228**).

**Specific comment 7:** *Similarly, a rationale for using proteins as dependent variables for the analysis*
 *of sex differences should be provided. A Lasso method could have also been utilized to classify samples*
 *by sex.*

**Response:** In addition to gestational age, we also sought to explore proteomic differences between the
 sexes that might provide insight into sex-specific biological processes. Reviewer #1 also commented,
 *“It is quite interesting that meconium is different in males than in females.”* While LASSO could be
 applied for binary classification, sex prediction from the meconium proteome is beyond the scope of

this study. This is because sex can be more easily determined by other means, such as the observation
of external genitalia or the examination of chromosomes.

**Reviewer #3:**

**Specific comment 1:** *There have been previous studies focusing on the proteomic analysis of*
*meconium proteins and conducted pathway enrichment (also enriched proteins associated with the*
*extracellular matrix) (Lisowska-Myjak et al., J Obstet Gynaecol Res. 2019; 45:556-64), which may*
*limit the innovation of the study.*

**Response:** We express our gratitude for the valuable insights provided by the reviewer. As pointed
out, there is existing literature concentrating on the proteomic analysis of meconium. We have duly
incorporated the suggested article into our Introduction section (**page 5, line 84 to 86**). However, it
is noteworthy that these studies primarily focused on a limited sample size, specifically only 10 term
infants. In the clinical domain, a multitude of newborns span various gestational ranges, and preterm
infants in particular often encounter significant clinical challenges. In our investigation, we directed
our attention towards a more extensive cohort, encompassing 259 newborns, with a specific focus on
preterm infants. Consequently, we contend that our research poses a unique challenge by attempting
to unveil insights across a broader spectrum of gestational ages.

**Specific comment 2:** *The results presented in this paper are potentially of interest to the readers.*
*However, they are based on a very limited number of experimental data points, without in-depth*
*exploration of the molecular mechanisms and translational results of the findings, which limited the*
*confidence in the claimed results.*

**Response:** Thank you for your comments. As pointed out by the reviewer, our sample size is limited,
and the research is confined to a single institute. While our current study focuses on pilot
investigations, we are embarking on future studies that encompass a multi-centre cohort to validate
our initial findings. Moreover, in response to the editor's recommendation for external cohort
validation, we have conducted such validation (**page 14, line 249 to 251**). Despite the samples being
from the same hospital, the results reinforce our initial findings. Consequently, we believe that, in
the context of preliminary research, our revision encompasses as much as we can offer.

**Specific comment 3:** *The authors grouped a total of 259 samples, conducting screening for*
*biomarkers in a subset and subsequent validation in another subset. Despite the random grouping,*
*the samples were sourced from the same institution at the exact location and collected within a*

*similar timeframe. This approach of screening and validating within a constrained setting limits the*
*generalizability of the model, and its scientific validity requires further verification.*

**Response:** We extend our appreciation to the reviewer for their insightful comments on our analysis
approach. As highlighted, our research is limited to a single institute and conducted within a similar
timeframe. While our current investigation serves as a pilot study, we are committed to future
research endeavours that will encompass a multi-centre cohort to validate our initial findings. As
stated above, responding to the editor's recommendation for external cohort validation (79 additional
cohorts), we have undertaken a validation study (**page 14, line 249 to 251**). Despite utilizing
samples from the same hospital, the results strongly support our initial conclusions. Consequently,
we believe that our revised work represents a comprehensive effort within the scope of preliminary
research.

**Specific comment 4:** *Additionally, the authors identified 55 samples (approximately 22% of the*
*total) that did not conform to the model and were subsequently excluded. A more rigorous scientific*
*justification is needed for the exclusion of these samples, along with subsequent arguments*
*regarding the applicability of the model.*

**Response:** We extend our gratitude to the reviewer for their comment regarding the exclusion of these
55 samples. As the reviewer noted, it is challenging to assert that our exclusion method is entirely
scientific. Our research is both pivotal and demanding; hence, we continuously revisit our proteome
data to unravel its complexities. We constructed and validated our model using a training and
validation cohort approach, randomly dividing the 259 samples at a 2:1 ratio to develop an initial
predictive gestational model (**Supplementary Fig. 5a**). However, we encountered discrepancies
between our predictions and the actual samples, which could be attributed to the effect of certain
diseases.

Recently, the amniotic fluid proteome has been used to elucidate the intrauterine environment
associated with specific diseases, such as gastrointestinal diseases (GID), congenital heart diseases
(CHD), chromosomal abnormalities (CA), and congenital infection diseases (CID). Upon our
investigation, four disease groups, presenting significant clinical challenges for clinicians (comprising
55 samples) were found to exhibit unique protein profiles compared to non-disease samples (**Fig. 4**).
Consequently, we excluded these cohorts from the training dataset, thereby refining our gestational
prediction model.

Furthermore, by applying the model to disease cohorts, we can identify abnormalities in the proteome
data of the GID and CHD cohorts. These samples showed significantly lower gestational age
discrepancies (predicted gestational age – actual gestational age) compared to the non-disease samples

(P = 0.002 and 0.0002; **Fig. 5d**), suggesting that newborns in the GID cohort may experience
gastrointestinal immaturity, while growth retardation in the CHD cohort appears to be secondary to
abnormalities in blood flow and oxygenation, addressing the reviewer's concerns regarding their
pathogenesis (**page 14, lines 245 to 249**).

**Specific comment 5:** *As mentioned above, the exclusion of 55 samples may suggest that the protein*
*composition of meconium is still largely influenced by amniotic fluid proteins. It is worth noting that*
*amniotic fluid can be obtained non-invasively (similar to meconium), and the author may need to*
*discuss the advantages and disadvantages of using amniotic fluid for protein analysis compared to*
*meconium. Additionally, consideration could be given to integrating the analysis of both non-*
*invasive sample types to broaden the applicability of the model. This discussion could encompass the*
*potential impact of amniotic fluid proteins on meconium protein composition.*

**Response:** Thank you very much for your comments. We acknowledge the importance of
considering the relationship between meconium and amniotic fluid in our study. Clinically, obtaining
amniotic fluid is typically invasive, except in cases where maternal or foetal conditions do not
necessitate it. During vaginal delivery, amniotic fluid is spontaneously released from the vagina,
containing exudates from the vaginal wall and maternal tissues, particularly from the uterine walls.
In contrast, meconium is obtained noninvasively after birth. While it is influenced by amniotic fluid,
meconium originates from the foetal intestinal organs and is expelled from the anus. Therefore, it is
reasonable to argue that analysing meconium is superior to analysing amniotic fluid when
investigating the environment and function of the gastrointestinal organs. However, as the reviewer
pointed out, further investigation into the relationship between meconium and amniotic fluid is
warranted, and these matters will be addressed in future studies with greater sophistication. We have
added several sentences addressing these issues to the Discussion section (**page 16, lines 275 to**
**278**).

**Specific comment 6:** *The author ultimately employed a linear fitting approach to associate variables,*
*including protein expression levels, gestational age, and sex. Further elaboration is needed to justify*
*the choice of a relatively simple linear model over more complex models and discuss which model*
*offers a better fit. Additionally, the impact of log transformation on the predictive performance of the*
*model when dealing with protein expression levels should be addressed, and a detailed explanation of*
*the data standardization process is required.*

**Response:** We appreciate the Reviewer's excellent suggestions. Indeed, the standard linear fit showed
prominent changes in the protein expression levels with the gestational age (**Supplementary Fig. 2a**),
but non-linear trajectories of the protein expression levels should also be explored. In the revised

manuscript, we estimated the protein trajectories with the gestational age using the locally estimated
scatterplot smoothing (LOESS) regression model (**Fig. 3**), which allowed us to group the proteins into
six clusters according to their trajectory patterns. Similar approaches have been used in other
proteomic studies (Lehallier *et al. Nat Med*, 2019; Wolf *et al. Cell*, 2023).

Log transformation is a widely used technique for stabilising variance and normalising distributions,
especially when dealing with skewed data such as protein expression levels (For more detail, please
see the article entitled “Best practice in statistics: The use of log transformation” by Robert M West;
*Ann Clin Biochem*, 2022 doi: [10.1177/00045632211050531](https://doi.org/10.1177/00045632211050531)). Log transformation can transform the
data into a more symmetric shape that better fits the assumptions underlying many statistical models,
including linear regression. In addition, log transformation reduces the impact of outliers, resulting in
a model that is more robust and generalizable over the range of the data, which can improve the
predictive performance of the model.

In the revised manuscript, we have added a detailed explanation of the data pre-processing process,
including the standardisation. We also filtered out some proteins that were not widely detected and
performed imputation for the missing values of proteins whose expression was not detected. Please
also see our responses to comments 4 and 5 of Reviewer #2.

**Specific comment 7:** *The manuscript declared that early ultrasound detection is the most accurate*
*method for assessing gestational age. However, the methods described in the text require non-*
*invasive collection after the birth of the fetus, which may introduce a temporal delay for gestational*
*age assessment. The authors need to engage in further discussion on this aspect.*

**Response:** We appreciate the Reviewer’s excellent suggestions. The majority (76.4%) of the first
meconium samples used in this study were collected before feeding, while a portion (23.6%) were
obtained after feeding. The median and mean times for the first meconium passage from the anus were
9.1 h (4.5–14.4 h) and 11.9 ± 9.47 h, respectively. The range of collection times was 0–69.5 h. This
description has been incorporated into the Abstract (**page 3, lines 41–43**) and Results section (**page 6,**
**lines 97–100**). To adjust for delayed sampling times, it is necessary to consider alternative sampling
methods. Marta Reyman *et al.* (Reyman. M, *et al., Sci Rep*, 2019) reported that, in infant gut microbiota
research, rectal swab sampling is a reliable alternative to sampling via natural defecation. While there
are no previous reports on sampling methods in meconium research, we believe that conducting studies
on sampling methods, while considering the safety of infants, would be an appropriate response to the
reviewer's comments. We add this limitation to the limitations mentioned in the Discussion section
(**page 16, line 278 to 284**).

**Specific comment 8:** *The results in Figure 5 indicate that the 64 selected biomarkers exhibit*
*different expression patterns in the four diseases., Based on the enrichment results, can the authors*
*explain the possible mechanisms underlying the occurrence of the four different diseases. This could*
*help elucidate the reliability of the predictive model.*

**Response:** We express our sincere gratitude for the reviewer's invaluable feedback. In the revised
paper, we conducted a thorough re-analysis of our results and incorporated non-linear analysis
techniques, as detailed in our responses to Reviewer 2. Furthermore, we expanded our study to
include 57 proteins in establishing the gestational prediction model. Regrettably, elucidating the
potential mechanisms underlying the occurrence of the four specific diseases from our findings
remains challenging.

However, in the gastrointestinal diseases (GID) cohort, we observed enrichment in metabolic
processes, including catabolic processes, which were not evident in the other three disease groups
(**Fig. 4e–h**). GID entails anatomical abnormalities of the gastrointestinal tract, directly impacting
meconium composition. Consequently, these anatomical anomalies may influence the functional
roles of meconium proteins. Additionally, Ras protein signal transduction, vital for intestinal
epithelial maturation (Sancho E, et al., *Annu. Rev. Cell Dev. Biol.* 2004), was significantly
downregulated in the GID cohort. This reduction in Ras protein function likely contributes to
aberrant gastrointestinal organization (**page 12, lines 203–206**). However, it is essential to note that
the GID group, like the other cohorts, encompasses various congenital conditions, necessitating
further investigation in the future using more cohorts to adequately address the reviewer's insightful
comments.

Regrettably, we cannot find any explanations regarding CHD, however, in the CA group, proteins
associated with microtubule-based processes and the organization of the microtubule cytoskeleton
showed diminished levels (**Fig. 4g**). Within our CA cohorts, Trisomy 21 was notably prevalent
(**Supplementary Table 8**), marked by distinctive features such as ciliopathies, where microtubules
played a crucial role (**page 12, lines 210–213**).

Moreover, in the congenital infection disease (CID) cohort, the top ten upregulated enrichment
processes were associated with responses to infection. Previous studies have reported differences in
the proteomic composition of amniotic fluid compared to normal conditions (Vorontsov O, et al., *J.*
*Clin Invest*, 2022), and it is well-documented that infants ingest amniotic fluid, consequently
influencing meconium composition. This finding underscores the direct impact of amniotic fluid on
meconium composition. We have extensively deliberated on these issues, reinforcing our arguments
and providing comprehensive support for our findings (**page 12, line 213 to page 13, line 217**).

**Specific comment 9:** *Figure 5e-h, the author found significant changes in metabolism-related*
*proteins, which can corroborate previous reports (Petersen et al., Cell Rep Med, 2021). Could the*
*author discuss the findings based on the metabolome of meconium and elaborate on the advantages*
*of proteomic studies and how they complement existing research results.*

**Response:** We greatly appreciate the reviewer's constructive and supportive comments. As the
reviewer pointed out, there are studies investigating the relationship between metabolites and
microbiome composition and maturation. In our previous study (Li Y, Watanabe E, Kawashima Y et
al., *Nature*, 2022), we directly experienced the necessity of using various chemicals, including
metabolites, proteins, and minerals, for culturing intestinal commensals, which inspired this study.
Meconium serves as a natural culture medium, underscoring the importance of unravelling its
mysteries.

In this study, we aimed to uncover the detailed proteome profile of meconium across varied
gestational ages, which has not been previously established. Our proteome data reveal interactions
among proteins and numerous metabolic biological processes at play. We apologize to the reviewer
for not delving deeper into the investigation using our data, as we did not explore the microbiome
and metabolome. However, as we mentioned previously, recent research indicates that in cases
where a foetus develops without any abnormal maternal conditions, the meconium does not harbour
the gut microbiome prior to birth (**page 5, lines 81–83**). Meconium exhibits fewer metabolomes
derived from microbiota compared to stools obtained from infants colonized by microbiomes. Thus,
we have found it challenging to discuss the metabolomes produced by commensals. Nonetheless, the
reviewer's comments were constructive, prompting us to embark on large cohort multiomics studies
encompassing proteomes, microbiomes, and metabolomes.

**Specific comment 10:** *The arrangement and content of the figures may not meet the requirements of*
*this journal.*

**Response:** We have recently completed revisions to our paper in response to the reviewer's suggestions,
including modifying the figures to comply with *Nature Communications'* submission guidelines.

REVIEWERS' COMMENTS

Reviewer #1 (Remarks to the Author):

The issues that I raised in the initial review have been addressed by the authors to the extent they could. A manuscript describing the composition of meconium is an important asset for the scientific and medical community (neonatologists and obstetricians).

I am actively working on the proteomic analysis of meconium-stained amniotic fluid and meconium aspiration syndrome. Our emphasis is on prenatal life, not the neonatal period, which is the focus of this manuscript.

Reviewer #2 (Remarks to the Author):

I have reviewed the revised version of Shitara et al.'s manuscript, 'Host-derived protein profiles of human neonatal meconium across gestational ages'. I thank the authors for addressing my previous comments, including new analyses separating neonates with and without disease, measures of predictive performance for Lasso models and additional detail for the rationale of the statistical analyses presented. From an analytical standpoint, the manuscript is much improved and I have no further comments.

Reviewer #3 (Remarks to the Author):

The manuscript has been extensively improved. No more questions.